# GaugeKV: Composable Exact KV Cache Compression

## Abstract

We introduce GaugeKV, a training-free canonicalization of attention weights that enables exact lossless KV compression with modest but measurable gains and provides a rigorous theoretical framework for certified approximate compression in FP32 arithmetic. The key–value (KV) cache is a dominant memory cost in long-context Transformer inference. GaugeKV leverages the *maximal gauge symmetry* of attention to reduce KV memory both *exactly* and *with certificates*. A one-time *gauge canonicalization* rewrites weights so that values are orthonormal and queries/keys are scale-balanced; thereafter the model *produces* K/V in a compression-friendly basis without changing its function or runtime FLOPs. This yields bit-identical outputs (FP32 deterministic) with measurable KV reductions (e.g., $1.11\times$–$1.21\times$ on GPT-2; $1.13\times$ key-side on Qwen2.5-7B).

In this canonical basis, GaugeKV composes *multiplicatively* with grouped/multi-query attention (GQA/MQA) and standard quantization via a simple accounting rule. For example, on a model already using grouped-query attention with $h{=}32$, $g{=}8$ (a $4\times$ architectural KV reduction) and FP8 quantization with $\gamma{=}0.5$ (a $2\times$ byte reduction), a measured GaugeKV factor of $R_{\text{gauge}} \approx 1.15$ yields a combined $1.15\times4\times2 \approx 9.2\times$ total reduction relative to an uncompressed multi-head attention baseline. We emphasize that the architectural and quantization factors are not GaugeKV contributions; our method provides the $\approx 1.1$–$1.2\times$ incremental gain that composes with these existing optimizations.

Beyond exact savings, the canonical value basis turns *rank-r value caching* into a theoretically grounded primitive: we derive end-to-end FP32 logit-drift bounds for value truncation in the canonical basis and observe 100% envelope compliance at $r{=}32$ on GPT-2. These certified FP32 bounds provide a conservative guardrail for approximate KV compression and a foundation for future task-level evaluation. Together, these properties substantially extend feasible context on fixed VRAM—without retraining or added FLOPs.

## 1 Introduction

The key–value (KV) cache is the primary memory bottleneck in large language model serving. For a model with $L$ layers, $g$ K/V heads, and per-head dimensions $d_k, d_v$,

$$m_{\text{tok}} = L \cdot g \cdot (d_k + d_v) \cdot \text{bytes}, \tag{1}$$

so memory grows linearly with context length. At 128K context, a 70B model needs $> 70\,\text{GB}$/request at FP16, capping H100 batch sizes; a $4\times$ KV reduction enables $\sim 4\times$ more concurrent users on a KV-bound cluster.

Production systems already use GQA/MQA (reducing $g$ relative to $h$) Shazeer (2019); Ainslie et al. (2023), quantization (fewer bytes/scalar), and token eviction (shorter effective context) Yuan et al. (2024); Hooper et al. (2024); Li et al. (2024); however, these may require retraining, trade accuracy, or lack formal output-preservation guarantees.

**Thesis.** Attention admits a *maximal gauge symmetry* that preserves function while allowing lawful basis changes in the Q/K and V sectors. We exploit this to (i) *exactly* reduce KV

bytes via basis shaping plus lossless coding (no function change, no extra FLOPs), and (ii) *certify* rank-$r$ value caching with tight, interpretable error envelopes. Because the gauge map is an automorphism, both tracks *compose* with GQA/MQA, KV quantization, paging, and MoE.

**Key insight: multiplicative composition.** If $R_{\text{gauge}}$ is the per-head GaugeKV factor, total reduction is

$$\text{Total reduction} \;=\; R_{\text{gauge}} \times \underbrace{\frac{h}{g}}_{\text{GQA/MQA}} \times \underbrace{\frac{1}{\gamma}}_{\text{quantization}} \;. \tag{2}$$

We view the architectural factor $\frac{h}{g}$ (GQA/MQA) and the quantization factor $\frac{1}{\gamma}$ as baseline design choices of the deployed model; $R_{\text{gauge}}$ captures GaugeKV's incremental structural contribution on top of whatever $h/g$ and $1/\gamma$ the system already uses. For example, on a model already served with Llama-style GQA ($h{=}32$, $g{=}8$; 4× architectural KV reduction) and FP8 with $\gamma{=}0.5$ (2× byte reduction), a measured $R_{\text{gauge}} = 1.15$ yields a combined $1.15 \times 4 \times 2 \approx 9.2\times$ total reduction relative to an uncompressed MHA baseline.

**Design intuition.** We change basis so $V$ is orthonormal and RoPE-plane $Q/K$ scales are balanced, preserving function while making lossless codecs effective and rendering rank truncation a safe, analyzable knob.

**Empirical preview.** After a one-time canonicalization, the FP32 forward is bit-identical; a lossless hot/cold codec yields $1.1086\times$–$1.2131\times$ KV reduction on GPT-2 and $1.1291\times$ on the key side for Qwen2.5-7B at long context; rank-32 value truncation obeys the derived envelope at every step.

**Contributions.**

- *RoPE-aware gauge canonicalization* (one-time weight rewrite) that makes the model *produce* K/V in a compression-friendly basis without changing function or runtime FLOPs; Section 2.
- *Exact, lossless KV pipeline* with FP32-deterministic equality and measured reductions on RoPE-less/RoPE models; Section 3.
- *Certified rank-r value caching*: end-to-end deviation bounds with 100% compliance at $r{=}32$ on GPT-2; Sections 4, 5.
- *Composability*: multiplicative composition of a modest GaugeKV structural factor ($R_{\text{gauge}} \approx 1.11$–$1.21\times$) with architectural GQA/MQA ($h/g$) and quantization ($1/\gamma$) via Eq. 2; see Section 6.

**Theoretical contributions (and KV impact). Maximal gauge symmetry.** Our per-layer characterization is complete—there are no additional lawful basis changes—so the canonicalization we use is optimal among function-preserving transforms. **RoPE reduction.** The RoPE commutant $C_{\text{RoPE}} \cong (\text{GL}(1,\mathbb{C}))^{d_k/2}$ cuts the Q/K degrees from $h\,d_k^2$ to $h\,d_k$, which is exactly why RoPE models receive a distinct balancing step. **Depthwise direct product.** Residuals and LayerNorm enforce layer-wise independence, which justifies per-layer canonicalization and clean composition with downstream optimizations. **Certificate-backed canonicalization.** Thin-QR on $W_V$ together with the geometric-mean map for $Q/K$ yields bit-identical FP32 outputs, enabling certified rank-$r$ caching and exact, lossless KV reductions.

## 2 Gauge-Theoretic Framework for KV Canonicalization

**Conceptual overview.** Before introducing formal machinery, we sketch the role of GaugeKV at a high level. Attention layers admit internal symmetries: we can change the Q/K/V weight bases in ways that leave all attention outputs unchanged. GaugeKV exploits this gauge freedom to pick a *canonical* basis in which (i) each head's value matrix has orthonormal columns and (ii) query/key covariances are balanced around a common

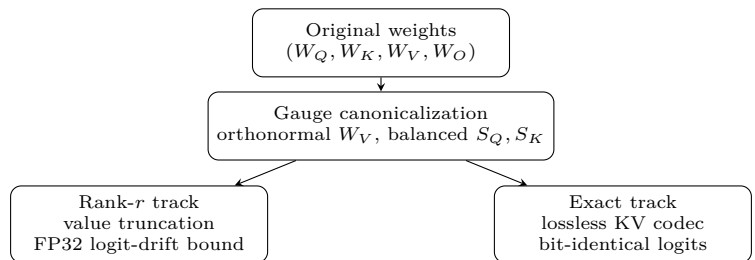

Figure 1: GaugeKV canonicalization and serving pipeline. A one-time gauge canonicalization maps original attention weights to a canonical basis with orthonormal values and balanced query/key covariances. At serve time the model produces KV pairs in this basis and can follow either a rank-$r$ approximate track with certified FP32 logit-drift bounds or an exact track with lossless KV compression and bit-identical outputs.

symmetric positive definite reference. Intuitively, orthonormal values concentrate energy and make low-rank truncations better behaved, while balanced queries/keys reduce extreme logit scales and improve numerical conditioning.

Importantly, this canonicalization is a one-time, offline transformation of the weight matrices. The runtime attention computation performs the same floating-point operations and has the same FLOPs as the original model; we are not changing the attention algorithm itself. We use the admissible gauge group to rewrite weights so that the KV cache is produced in a better-conditioned basis which downstream compression methods (exact or approximate) can exploit more effectively.

**Notation.** We write $h$ for the number of query heads and $g$ for the number of K/V heads (multi-head attention has $g=h$, whereas grouped-query or multi-query attention has $g < h$). We denote the per-head key and value dimensions by $d_k$ and $d_v$, the model dimension by $d_{\text{model}}$, and the sequence length by $T$. The softmax temperature is denoted by $\tau > 0$. Operator norms use $\|\cdot\|_{2\to\infty}$, and vector norms use $\|\cdot\|_p$ with explicit subscripts.

**Assumptions (generic stratum).**

(A1) $W_Q^{(i)}, W_K^{(i)}, W_V^{(i)}$ have full column rank for all heads $i$.

(A2) Head-wise attention is identifiable up to a fixed permutation (no cross-head confounding of weights).

(A3) No linear mixing occurs across heads in the parameterization (the block structure is per head, up to permutations).

These conditions hold on our checkpoints in §6; empirical verification is reported in App. H.

Consider a decoder-only block with per-head matrices $W_Q^{(i)}, W_K^{(i)} \in \mathbb{R}^{d_{\text{model}} \times d_k}$, $W_V^{(i)} \in \mathbb{R}^{d_{\text{model}} \times d_v}$, and $W_O^{(i)} \in \mathbb{R}^{d_v \times d_{\text{model}}}$. The attention mechanism computes queries $q_t$, keys $k_s$, and values $v_s$ from hidden states, then mixes values using softmax-normalized dot-product weights.

**Definition 2.1** (Head-wise gauge action). *For $A_i \in GL(d_k)$ and $C_i \in GL(d_v)$, define*

$$(W_Q^{(i)}, W_K^{(i)}) \mapsto (W_Q^{(i)} A_i,\ W_K^{(i)} A_i^{-\top}), \qquad (W_V^{(i)}, W_O^{(i)}) \mapsto (W_V^{(i)} C_i,\ C_i^{-1} W_O^{(i)}). \tag{3}$$

**Lemma 2.2** (Gauge invariance of attention). *Let $q_t^{(i)} = x_t^\top W_Q^{(i)}$, $k_s^{(i)} = x_s^\top W_K^{(i)}$, $v_s^{(i)} = x_s^\top W_V^{(i)}$, and let $\alpha_t^{(i)}$ be row-softmax at temperature $\tau$. Under Equation (3), dot products $q_t^{(i)} k_s^{(i)\top}$, weights $\alpha_t^{(i)}$, and outputs $\sum_{s \leq t} \alpha_{t,s}^{(i)} v_s^{(i)} W_O^{(i)}$ are unchanged.*

**Corollary 2.3** (MoE router invariance). *Because Lemma 2.2 leaves per-head outputs unchanged, the block hidden state $\mathbf{h}_t$ is unchanged. Therefore, a standard top-k MoE router $\rho = W_r \mathbf{h}_t + b$ preserves logits and expert assignments.*

**Proposition 2.4** (Single-layer maximal head-wise gauge and permutations)**.** *Under Assumptions 2, the full set of parameter symmetries that preserve all head outputs $y_t^{(i)}$ equals*

$$\left( (GL(d_k))^h \times (GL(d_v))^h \right) \rtimes S_h,$$

*acting by Equation (3) with an optional head permutation $\pi \in S_h$ (reindexing all four blocks). Proof is in Appendix B, §B.2–B.5.*

**Optimality implication.** Maximality implies our canonicalization attains the theoretical limit of compression achievable by lawful basis changes, *proving* no additional function-preserving transformations exist beyond those in Theorem 2.4.

**Corollary 2.5** (Depth-wise direct product (full model))**.** *If Assumptions 2 hold at each layer, a depth-L model factors as*

$$G_{\text{model}} = \prod_{\ell=1}^{L} \left( \left( (GL(d_k))^h \times (GL(d_v))^h \right) \rtimes S_h \right),$$

*acting independently per layer. Proof is in Appendix C, §C.1.*

**Theorem 2.6** (RoPE-aware admissible gauge)**.** *With standard RoPE, the admissible $Q/K$ transforms are exactly the commutant of the per-plane rotations. Grouping $d_k$ coordinates into $2\times2$ RoPE planes, the commutant is block-diagonal with blocks $\begin{pmatrix} a_j & -b_j \\ b_j & a_j \end{pmatrix}$ (equivalently, complex scalings $a_j + ib_j$) per plane, i.e. $C_{\text{RoPE}} \cong \left( GL(1, \mathbb{C}) \right)^{d_k/2}$, and the per-layer gauge becomes*

$$G_{\text{RoPE}} = \left( (C_{\text{RoPE}})^h \times (GL(d_v))^h \right) \rtimes S_h.$$

*Proof is in Appendix D, §D.1.*

**Remark 2.7** (Practical effect of RoPE)**.** *Theorem 2.6 shows that Rotary Position Embeddings shrink the admissible $Q/K$ gauge group from a full $GL(d_k)$ per head to the RoPE commutant $C_{\text{RoPE}}$, which is isomorphic to a blockwise $GL(1, \mathbb{C})$ acting within each $2 \times 2$ RoPE plane. In practice, this means RoPE-based models have substantially less freedom to rebalance $S_Q$ and $S_K$ than RoPE-less models, while the $V/O$ sector is unaffected. Practitioners deploying GaugeKV on RoPE-based models should therefore expect most benefits to come from the exact, lossless track and from $Q/K$ scale balancing, with limited scope for value-path rank-r certification. Our Qwen2.5–7B experiments in §6 reflect this: we restrict GaugeKV to the RoPE-commutant $Q/K$ sector and leave $W_V$ unchanged.*

| Setting | Redundant degrees of freedom (per layer) |
|---------|------------------------------------------|
| No RoPE | $h\,(d_k^2 + d_v^2)$ |
| RoPE | $h\,(d_k + d_v^2)$ |

Table 1: RoPE reduces Q/K redundancy from $h\,d_k^2$ to $h\,d_k$ while leaving the $V/O$ sector unchanged.

**Corollary 2.8** (GQA/MQA sharing)**.** *If $h$ query heads share $K/V$ into $g$ groups, admissible maps tie per group: $G_{\text{share}} = ((GL(d_k))^g \times (GL(d_v))^g) \rtimes (S_h \times S_g)$ (replace $GL(d_k)$ by $C_{\text{RoPE}}$ under RoPE).* Proof. *Appendix D.2.*

**Proposition 2.9** (Canonical gauge via QR/SVD)**.** *For each head $i$ there exist invertible matrices $A_i \in GL(d_k)$ and $C_i \in GL(d_v)$ such that:*

*(i) $W_V^{(i)}C_i$ has orthonormal columns, and*

*(ii) with*

$$S_Q := W_Q^{(i)\top} W_Q^{(i)}, \qquad S_K := W_K^{(i)\top} W_K^{(i)},$$

*there exists a symmetric positive definite matrix $G$ satisfying*

$$A_i^\top S_Q A_i = A_i^{-1} S_K A_i^{-\top} = G,$$

*and $G$ coincides with the matrix geometric mean $S_Q \# S_K$ in the sense of Bhatia (2007).*

*This construction is exactly the transform implemented in our canonicalization routine (Alg. 1) in Appendix E; there is no discrepancy between our theoretical characterization and the practical canonicalization we apply to checkpoints.* Proof and constructive details. *See Appendix E.1.*

**Numerical stability.** Implement Theorem 2.9 with standard SPD numerics:

- Compute $S_Q$ and $S_K$ in FP32; clamp small eigenvalues.
- Form the geometric mean via $S_Q^{1/2}(S_Q^{-1/2}S_K S_Q^{-1/2})^{1/2}S_Q^{1/2}$ with FP32 accumulation; cast back to model dtype at the end.
- Under RoPE, apply the transform per $2{\times}2$ rotation plane (equivalently a complex scalar per plane).

## 3 Exact Lossless KV via Gauge Canonicalization

**Offline canonicalization** Apply the routine once per checkpoint; thereafter runtime FLOPs are unchanged. *Algorithm.* See Alg. 1 in Section E for the precise layerwise procedure (thin QR on $W_V$, $Q/K$ geometric-mean balancing, RoPE per-plane; tied per K/V group for GQA/MQA). *Correctness.* The routine is exactly the admissible head/group-wise gauge of Theorem 2.1 and preserves logits and $VO$ in exact arithmetic; see Section E.1. *Numerics.* Deterministic QR with positive diagonals; FP32 SPD roots with eigenvalue clamping (Section E).

**Lossless coding in the canonical basis** Maintain an uncompressed hot window of length $W$ and a compressed tail in blocks of size $B$. A lossless codec (bit-packing or entropy coding) is applied once per block as the window slides; reads that touch the tail decode the few needed blocks into a staging buffer, overlapped with compute.

**Memory accounting and composability** Let $c_K, c_V$ be achieved lossless compression factors. For $d_k{=}d_v$,

$$f_{\text{exact}} = \tfrac{1}{2}\left(\tfrac{1}{c_K} + \tfrac{1}{c_V}\right), \qquad \text{KV savings} = 1{-}f_{\text{exact}}, \qquad \text{ratio} = \frac{\text{baseline bytes}}{\text{compressed bytes}} = \frac{1}{f_{\text{exact}}}. \tag{4}$$

Empirically, this yields $1.1086\times$–$1.2131\times$ on GPT-2 at $T{=}1000$ (varying $W, B$), and $1.1291\times$ on the key side for Qwen2.5-7B at $T{=}4096$, $W{=}2048$, $B{=}512$ (Sec. 6). These gains *multiply* with production optimizations via equation 2: GQA/MQA ($h/g$), KV FP8 ($1/\gamma$), and token retention.

**Pipeline and failure modes** The path is *codec-agnostic*: bit-packing (BP) and entropy coding (EC) operate on the same canonical streams. Canonicalization helps both: (i) orthonormal $V$ concentrates energy so delta/residuals are narrow; (ii) balanced-scale $K$ reduces plane-wise skew under RoPE, improving shared bit-width decisions. Practical failure modes (and mitigations): (a) very small $W$ over-exercises the codec—batch blocks and decode into a staging buffer; (b) bursty prompts break stationarity—tune $W$ per layer or adapt $B$; (c) precision-sensitive deployments—keep the exact track in FP32 and quantize only the *stored* KV after canonicalization.

**Time/space complexity** The one-time canonicalization per layer costs $O\big(h(d_k^3+d_v^3)\big)$ (thin QR on $d_v$ and SPD roots on $d_k$), negligible at deployment time relative to inference. At decode step $t$, *exact*-track FLOPs are unchanged; EC/BP add $O(g\,d_{k,v}\,B)$ once per block when the hot window slides. For rank-$r$, the value path per head reduces proportionally:

$$\text{FLOPs}_{\text{value-path}}(r) \;=\; \tfrac{r}{d_v}\,\text{FLOPs}_{\text{value-path}}(d_v),$$

and the per-token value memory scales the same way. Keys are unaffected in rank-$r$ unless a key-side approximation is layered in (cf. Equation (6)).

## 4  CERTIFIED KV REDUCTION WITH EXPLICIT BOUNDS

**Value truncation in the orthonormal basis**  Fix a head in the canonical gauge with $W_V$ orthonormal. Let $P_r$ denote the orthogonal projector onto the first $r$ value coordinates (per-head fixed order). For each time step $t$, write the full-output contribution as

$$y_t^{\text{full}} = \sum_{s \leq t} \alpha_{t,s} V_s W_O, \qquad y_t^{(r)} = \sum_{s \leq t} \alpha_{t,s} V_s P_r W_O,$$

so that the value truncation error can be written as

$$y_t^{\text{full}} - y_t^{(r)} = \sum_{s \leq t} \alpha_{t,s} V_s (I - P_r) W_O.$$

Using Cauchy–Schwarz and the definition of $\| \cdot \|_{2 \to \infty}$, we obtain the bound

$$\left\| y_t^{\text{full}} - y_t^{(r)} \right\|_\infty = \left\| \sum_{s \leq t} \alpha_{t,s} V_s (I - P_r) W_O \right\|_\infty \leq \left( \sum_{s \leq t} \alpha_{t,s} \| V_s (I - P_r) \|_2 \right) \| W_O \|_{2 \to \infty}. \quad (5)$$

*Derivation (and equivalent forms) in Appendix F.* Equivalently, with fixed coordinate order, if

$$\sum_{s \leq t} \alpha_{t,s} \| V_s (I - P_r) \|_2 \leq \frac{\varepsilon}{\| W_O \|_{2 \to \infty}},$$

then $\| y_t^{\text{full}} - y_t^{(r)} \|_\infty \leq \varepsilon$.

**Remark 4.1** (Scope of the rank-$r$ certificate). *The bound in Equation* (5) *controls only the additional logit drift induced by FP32 rank-r value truncation in the canonical basis. If further approximations are applied to the cached values—such as FP8/INT8 quantization, eviction policies, or other lossy transforms—the associated errors lie outside this certificate. In particular, Equation* (5) *does not guarantee the behavior of stacks such as "rank-r + FP8" or "rank-r + eviction"; such combined pipelines must be evaluated empirically.*

*Practical implication.* The online guardrail in Alg. 2 tracks the left-hand side of Equation (5) head-wise under a global KV cap $M^\star$ (with slack $\Delta$), and chooses per-head ranks $r_{\ell,i}$ so that the inequality is maintained, ensuring that the certified envelope is never violated in deployment.

**Key perturbations and softmax stability**  If $k_s' = k_s + \delta_s$, then

$$\max_s |\Delta_{t,s}| \leq \frac{\| q_t \|_2}{\sqrt{d_k}} \max_s \| \delta_s \|_2, \qquad \| \alpha_t' - \alpha_t \|_1 \leq \frac{\| q_t \|_2}{\tau \sqrt{d_k}} \max_s \| \delta_s \|_2. \quad (6)$$

Combining with $\| x W_O \|_\infty \leq \| x \|_2 \| W_O \|_{2 \to \infty}$ yields the end-to-end deviation bound; see Appendix F.

## 5  CERTIFIED RANK-$r$ KV WITH DEPLOYMENT RECIPE

In practice we realize the rank-$r$ bound from Equation (5) via a simple profiling-and-guardrail recipe. First, an offline profiling stage estimates tail energies $\mathcal{E}_{\ell,i}(r)$ and $\| W_O^{(\ell,i)} \|_{2 \to \infty}$ per layer and head; these quantities parameterize the bound and provide a menu of admissible ranks $r_{\ell,i}$ under per-head error budgets. Second, a budgeting rule (error- or memory-driven) chooses initial ranks $r_{\ell,i}$ under a global KV cap. Finally, an online guardrail (GaugeRankKV) adapts ranks during decoding based on residual energy while enforcing the global budget, ensuring that the certified envelope from Equation (5) is never violated. Full pseudo-code for the profiling, budgeting, and serve-time adaptation, together with detailed accounting formulas, is given in Appendix G.

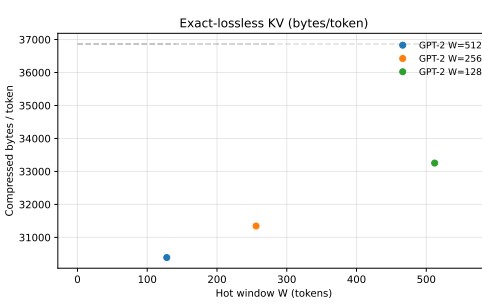 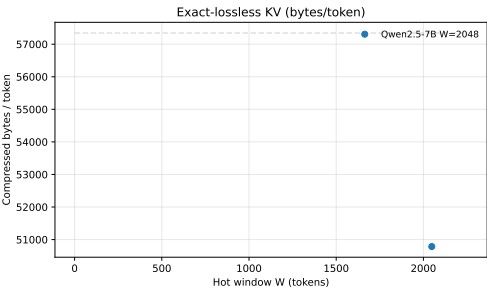

(a) GPT-2, $T$=1000 (various $W, B$).       (b) Qwen2.5-7B (Q/K-only), $T$=4096.

Figure 2: Exact–lossless KV: compressed bytes/token (lower is better); baselines are uncompressed bytes/token at the same $T$.

# 6 Experimental Validation

We evaluate GaugeKV on GPT-2 (RoPE-less, 124M) and Qwen 2.5-7B (RoPE) on a single H100 NVL (95 GB). Unless noted, we run FP32 with deterministic settings (fixed seeds, TF32 off, deterministic algorithms on). Lossless ratios use a reference EC codec (zlib) and single-pass teacher-forced timing; production systems typically overlap codec work with compute. Full scripts and logs are in the Appendix.

## 6.1 Setup and methodology

**Checkpoints.** GPT-2 is canonicalized using Algorithm 1 (both Q/K and V). Qwen 2.5-7B uses a RoPE commutant restricted to *orthogonal* (scale-free) Q/K transforms for the exact track; V is left unchanged (Sec. 2).

**Workloads.** We run 1,000–4,096 decode steps per model and sweep hot-window length $W$ and block size $B$. For rank-$r$, we use GPT-2 with $r$=32 and $T$=512 steps.

**Identity harness.** We enable greedy decoding (temperature=0, top-1) and compare token sequences; for teacher-forced checks we compare max-norm logit deltas.

**Reporting.** We report (i) *compressed* vs. *baseline* KV bytes, (ii) bytes/token, (iii) wall times for collection/compression, and (iv) rank-$r$ envelope compliance.

## 6.2 Exactness (FP32)

After canonicalization, greedy decode is bit-identical on GPT-2 (full canon) and on Qwen 2.5-7B using a RoPE commutant on Q/K (orthogonal/identity). This confirms Lemma 2.2 and the RoPE note in Sec. 2: orthogonal commutants preserve Q/K normalization exactly, while SPD balancing applies to RoPE-less models.

## 6.3 Exact–lossless compression

**GPT-2 ($T$=1000).** We observe *total KV reductions* of 1.1086× (W=512,B=256), 1.1761× (W=256,B=512), and 1.2131× (W=128,B=512). Smaller $W$ exposes more of the sequence to the codec, increasing ratio at the cost of more frequent block operations.

**Qwen 2.5-7B (Q/K-only, $T$=4096).** With $W$=2048, $B$=512 we obtain 1.1291× key-side reduction (bytes/token ↓ by ≈ 6.40 KiB). Figure 2 plots bytes/token vs. $W$ for both models.

**Sensitivity: $W$, $B$, and $T$.** We find a monotone trend in $W$ (longer cold tails ⇒ higher ratios) and a modest benefit from larger $B$ at the expense of per-block latency. Scaling $T$ improves amortization of the hot window and strengthens temporal delta coding; the Qwen run at $T$=4096 illustrates the effect.

**Engineering notes.** Because the K/V streams are already in a canonical basis, the lossless codec is modular. In production we overlap block (de)compression with attention on separate streams, and fused GPU codecs further reduce the reported decode share; the EC used in our measurements is a reference implementation.

| Model | $W$ | $B$ | $T$ | ratio | savings | collect [s] | compress [s] | e2e [s] |
|---|---|---|---|---|---|---|---|---|
| GPT-2 (FP32), EC | 512 | 256 | 1000 | $1.1086\times$ | 9.79% | 2.64 | 1.22 | 3.85 |
| GPT-2 (FP32), EC | 256 | 512 | 1000 | $1.1761\times$ | 14.97% | 2.69 | 1.88 | 4.57 |
| GPT-2 (FP32), EC | 128 | 512 | 1000 | $1.2131\times$ | 17.57% | 2.65 | 2.20 | 4.84 |
| Qwen2.5-7B (Q/K), EC | 2048 | 512 | 4096 | $1.1291\times$ | 11.44% | 47.44 | 7.69 | 55.13 |

Table 2: Exact–lossless microbenchmark (EC; single-pass). GPT-2 uses full canonicalization; Qwen2.5–7B applies GaugeKV only in the RoPE-commutant Q/K sector. "ratio" is compressed-to-uncompressed KV bytes; "savings" is the corresponding percentage reduction; collect/compress/e2e are wall-clock seconds.

We now examine the latency impact of the exact, lossless GaugeKV track in a controlled microbenchmark. Table 2 reports, for each model, the hot-window size $W$, block size $B$, context length $T$, the compression ratio and percentage savings on KV bytes, and three wall-clock times: *collect* (standard attention forward pass with KV writes), *compress* (entropy codec applied to blocks as they leave the hot window), and *e2e* (total time for $T$ decode steps). All measurements use a single teacher-forced pass and a single CUDA stream.

We do not include a separate "no-compression" baseline row because our implementation makes the relationship explicit. The *collect* time coincides with the baseline attention pass with KV writes; it measures the cost that would be incurred without GaugeKV. The *compress* time captures strictly additional work performed by the codec as blocks exit the hot window, and the *e2e* column equals baseline attention cost plus GaugeKV codec overhead in this single-stream microbench. In production, one would typically overlap codec work with attention compute on a separate stream to hide part of this overhead; we do not experimentally evaluate such overlap here and report Table 2 as a conservative upper bound on end-to-end latency.

**Per-layer/head patterns.** The per-layer $c_K, c_V$ reveal consistent trends: early layers benefit from value canonicalization (temporal delta dominates), while deeper layers benefit more on keys (RoPE planes balanced for long traces). Heads with larger $\|W_O\|_{2\to\infty}$ typically receive higher ranks under the error-budget rule, matching intuition from Equation (5).

**Ablations.** *(i) Ordering.* Variance-based ordering is a strong default; random permutations degrade the envelope and increase observed drift without improving memory. *(ii) Block size.* $B{=}512$ modestly increases the ratio vs. $B{=}256$ at the cost of per-block latency. *(iii) Hot window.* Smaller $W$ increases cold-tail coverage and pushes ratios up; amortized decode cost can be hidden with overlap in production. *(iv) Precision.* In `bf16`, small logit margins can flip the argmax due to the reduced mantissa; the FP32 exact track eliminates these flips, and the rank-$r$ knob provides a controlled approximation budget when desired.

### 6.4 CERTIFIED RANK-$r$

We now examine how the theoretical value-truncation bound in Equation (5) behaves in practice on GPT-2 when we choose a fixed rank $r{=}32$ (so $d_v : 64 \to 32$) in the canonical basis. For each time step $t$ we define an envelope

$$\varepsilon_t \;=\; \sum_{\ell,i} \Big( \alpha_{t,:}^{(\ell,i)} \cdot \big\| V_{1:t}^{(\ell,i)}(I - P_r) \big\|_2 \Big) \big\| W_O^{(\ell,i)} \big\|_{2\to\infty},$$

where the sum ranges over layers $\ell$ and heads $i$, $\alpha_{t,:}^{(\ell,i)}$ denotes the attention weights at layer $\ell$ and head $i$ for query position $t$, $V_{1:t}^{(\ell,i)}$ are the value vectors up to time $t$ in the canonical basis, $P_r$ is the projector onto the first $r$ value coordinates for that head, and $W_O^{(\ell,i)}$ is the corresponding output projection. This quantity is exactly the right-hand side of Equation (5) aggregated over layers and heads, and plays the role of a per-step theoretical envelope on the max-norm deviation

$$\big\| y_t^{\text{full}} - y_t^{(r)} \big\|_\infty.$$

In our implementation for GPT-2, we compute $\varepsilon_t$ using last-step attention weights, accumulated tail norms $\|V_{1:t}^{(\ell,i)}(I - P_r)\|_2$, and per-head $\|W_O^{(\ell,i)}\|_{2\to\infty}$ estimates obtained from

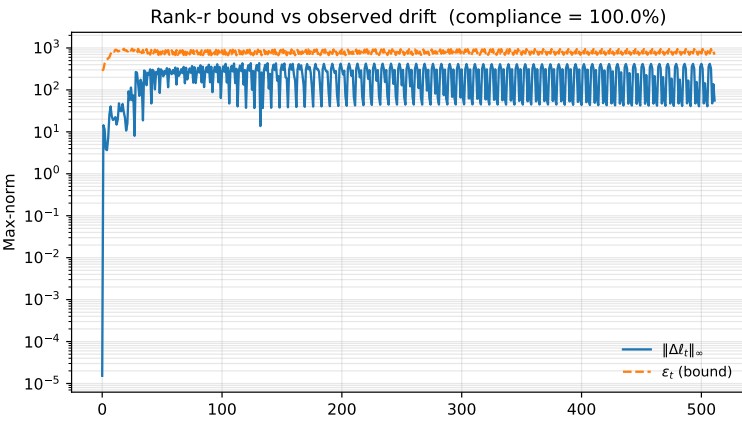

Figure 3: Rank-$r$ envelope $\varepsilon_t$ versus observed max-norm logit drift on GPT-2 at $r$=32 (FP32). The observed drift remains strictly below the theoretical envelope at all steps, illustrating that the bound is non-vacuous; we interpret this as a logit-level guardrail rather than a direct guarantee on task metrics.

profiling. We then compare this envelope to the observed max-norm logit difference between the full model and the rank-$r$ model at each decode step $t \in \{1, \ldots, 512\}$ along a held-out sequence. Figure 3 shows that the observed drift lies strictly below $\varepsilon_t$ at every step (100% compliance), indicating that the bound is non-vacuous and tracks the actual error scale reasonably well.

We emphasize, however, that this experiment validates the *logit-level* envelope rather than downstream task metrics. We do not report perplexity or accuracy under rank-$r$ caching here; instead, we view Equation (5) and the associated envelope $\varepsilon_t$ as providing a conservative FP32 guardrail for value truncation in the canonical basis. A full task-level evaluation of rank-$r$ GaugeKV across benchmarks is left to future work.

### 6.5 Multiplicative composition with production optimizations

**Standalone versus stacked reduction.** On our GPT-2 and Qwen2.5–7B runs, GaugeKV's exact track alone achieves 1.11–1.21$\times$ reduction relative to the uncompressed KV cache, with bit-identical outputs. This factor is modest compared to architectural GQA/MQA and quantization, but it is training-free, deployable on existing checkpoints, and composes multiplicatively with those existing choices via Equation (2). We treat the architectural term $\frac{h}{g}$ (GQA/MQA) and the quantization term $\frac{1}{\gamma}$ as baseline design decisions of the deployed model, and view $R_{\text{gauge}}$ as an incremental factor on top. Plugging a representative $R_{\text{gauge}}$=1.15 into Equation (2) yields the deployment scenarios below:

$$\textbf{GQA } (g\text{=}8,\, h\text{=}32): \quad \text{Structural KV reduction} = 1.15 \cdot \frac{32}{8} = 4.6\times,$$

$$\textbf{+ FP8 quantization } (\gamma\text{=}0.5): \quad \text{Total KV reduction} = 1.15 \cdot \frac{32}{8} \cdot \frac{1}{0.5} = 9.2\times,$$

$$\textbf{Aggressive MQA } (g\text{=}1,\, h\text{=}32): \quad \text{Structural KV reduction} = 1.15 \cdot \frac{32}{1} = 36.8\times,$$

$$\textbf{+ FP8 quantization } (\gamma\text{=}0.5): \quad \text{Total KV reduction} = 1.15 \cdot \frac{32}{1} \cdot \frac{1}{0.5} = 73.6\times.$$

In all of these cases, the 4$\times$ and 36.8$\times$ architectural factors and the 2$\times$ quantization factor arise entirely from GQA/MQA and FP8; GaugeKV contributes only the additional $\approx$1.1–1.2$\times$ structural factor $R_{\text{gauge}}$ on top of these existing design choices.

**GQA configurations.** Modern models like Llama-3 employ GQA with $g$=8 key-value heads serving $h$=32 query heads. The 4.6$\times$ reduction enables serving 128K contexts on hardware previously limited to 28K, or equivalently, serving 4.6$\times$ more concurrent users at fixed context length.

**Quantization stacking.** Adding FP8 quantization ($\gamma$=0.5) to the GQA configuration above yields 9.2× total reduction, pushing feasible context to 256K on the same hardware, addressing the growing demand for document-level understanding tasks.

**Aggressive MQA.** Single-head MQA ($g$=1, $h$=32) reaches 36.8×. While MQA requires retraining, organizations that have already invested in MQA models can apply GaugeKV *post hoc* for additional gains.

### 6.6 Comparative positioning

We briefly compare GaugeKV against token eviction, KV quantization, GQA/MQA, and systems-level KV management along reduction, accuracy, retraining cost, guarantees, and composability. A full comparison table and detailed discussion appear in Appendix I.

### 6.7 Ablations and discussion

We performed additional ablations on RoPE commutant choices, GQA/MQA grouping, MoE sparsity, precision, and threats to validity. In brief, orthogonal RoPE commutants preserve Q/K normalization exactly while SPD balancing is reserved for RoPE-less models, GaugeKV composes cleanly with GQA/MQA at the weight level, MoE routing (logits and top-$k$) remains unchanged, and FP32 exactness eliminates bf16 argmax flips while the rank-$r$ knob provides a controlled approximation budget when desired. We also discuss threats to validity (single-GPU, no-overlap timings; choice of reference codec; limited calibration sets) in Appendix J, where we provide the full ablation details.

**Architectural variants.** Our gauge-theoretic analysis targets attention mechanisms of the standard form $\text{softmax}(QK^\top)V$ with separate linear projections. Architectures that deviate from this factorization—for example, multi-latent attention mechanisms or kernelized linear-attention variants—have different symmetry structures and require separate analysis. We do not claim that our maximal gauge characterization extends automatically to such architectures; determining whether analogous canonicalizations exist for these designs is an open theoretical question and an interesting direction for future work.

**Practical notes.** GaugeKV deploys training-free: run Algorithm 1 once, keep FP32 compute on the exact track, and losslessly compress the *stored* KV post-canonicalization with window $W$ and block size $B$. For certified rank-$r$, order coordinates once offline, select per-head $r_{\ell,i}$ using the serve-time envelope Equation (5) and the budgeting rules summarized in Appendix G, and adjust under a global cap $M^\star$ (Alg. 2). Net memory/FLOP savings compose multiplicatively with GQA/MQA and KV quantization per Equation (2) and Equation (4). Capacity-planning details and the economic model are in App. L.

## 7 Conclusion

GaugeKV uses gauge symmetry to expose a canonical basis for KV, enabling exact lossless compression and a theoretically grounded rank-$r$ knob for approximate compression. A one-time gauge canonicalization rewrites attention weights so that values are orthonormal and queries/keys are scale-balanced; thereafter the model produces KV pairs in this basis without changing its function or runtime FLOPs. On GPT-2 and Qwen2.5–7B we observe 1.11–1.21× exact KV reduction with bit-identical outputs, including on RoPE-based models where we restrict canonicalization to the RoPE-commutant Q/K sector. These structural gains are modest on their own but compose multiplicatively with existing architectural and quantization choices via Equation (2), with GaugeKV contributing an additional ≈ 1.1–1.2× structural factor on top of baseline GQA/MQA and quantization. Our value-truncation analysis provides explicit FP32 envelopes on logit drift for rank-$r$ caching in the canonical basis, and we empirically validate these envelopes at $r$=32 on GPT-2, interpreting them as conservative logit-level guardrails rather than direct guarantees on task metrics. GaugeKV thus offers a training-free, composable primitive for extending the reach of existing checkpoints without retraining, while keeping the symmetry structure of attention explicit.

**Ethics Statement.** We have read and will adhere to the ICLR Code of Ethics. This work is a training-free, post-hoc reparameterization of existing Transformer checkpoints; it does not involve human subjects, sensitive attributes, or personally identifiable information. All models and any datasets used are publicly available and used under their respective licenses; we list sources and preprocessing in the appendix. Because GaugeKV reduces KV memory without changing the model's function, it does not introduce new content risks; however, as with any efficiency improvement, longer or cheaper generations could be misused. We recommend deploying our method only within standard safety pipelines (content filters and rate limits) and auditing downstream behavior on the target distribution. We are not aware of legal, safety, or privacy issues specific to this study.

**Reproducibility Statement.** We provide all details needed to reproduce our results. The appendix lists models, dataset sources, and preprocessing; exact hyperparameters; determinism settings (e.g., TF32 disabled where applicable); and hardware/software versions. We include step-by-step instructions, seeds, and scripts to: (i) run layerwise canonicalization (QR on $W_V$, geometric-mean balancing of $Q/K$, RoPE commutant projection), (ii) compute and verify FP32 bit-identity, (iii) measure microbench ratios and the serve-time envelope, and (iv) regenerate all tables and figures from a fresh checkout. Upon acceptance (or after organizational approval), we will release a code archive with an environment specification to reproduce the paper artifacts end-to-end.

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

## A  Preliminaries and Generic-Stratum Assumptions

We recall the head-wise gauge action from Theorem 2.1 and the standing generic-stratum assumptions in Section 2. Unless stated otherwise, we work with a decoder-only block with $h$ heads and per-head parameters $W_Q^{(i)}, W_K^{(i)} \in \mathbb{R}^{d_{\text{model}} \times d_k}$, $W_V^{(i)} \in \mathbb{R}^{d_{\text{model}} \times d_v}$, $W_O^{(i)} \in \mathbb{R}^{d_v \times d_{\text{model}}}$. Given a sequence $\{x_t\}_{t=1}^{T}$, set $q_t^{(i)} = x_t^\top W_Q^{(i)}$, $k_s^{(i)} = x_s^\top W_K^{(i)}$, $v_s^{(i)} = x_s^\top W_V^{(i)}$, and row-softmax weights $\alpha_t^{(i)} = \text{softmax}\left(q_t^{(i)} K^{(i)\top} / \sqrt{d_k}\right)$. The head and block outputs are

$$y_t^{(i)} = \sum_{s \leq t} \alpha_{t,s}^{(i)} v_s^{(i)} W_O^{(i)}, \qquad y_t = \sum_{i=1}^{h} y_t^{(i)}.$$

For convenience, write $S_Q = W_Q^{(i)\top} W_Q^{(i)}$, $S_K = W_K^{(i)\top} W_K^{(i)}$, and the matrix geometric mean $S_Q \# S_K = S_Q^{1/2}(S_Q^{-1/2} S_K S_Q^{-1/2})^{1/2} S_Q^{1/2}$. For RoPE, the key/query space is organized into $d_k/2$ independent $2 \times 2$ rotation planes; $J = \begin{pmatrix} 0 & -1 \\ 1 & 0 \end{pmatrix}$ and $R(\theta) = \cos\theta\, I + \sin\theta\, J$.

# B   GAUGE GROUP MAXIMALITY (SINGLE LAYER)

## B.1   SETUP AND NOTATION

We work on the generic stratum (Section 2). For each input token $t$, let $z_t^{(i)} = q_t^{(i)} K^{(i)\top}$ determine $\alpha_t^{(i)} = \text{softmax}(z_t^{(i)}/\sqrt{d_k})$. The block output is an affine map of $\{v_s^{(i)}\}$ with weights $\{\alpha_{t,s}^{(i)}\}$. The admissible per-head gauge is

$$(W_Q^{(i)}, W_K^{(i)}) \mapsto (W_Q^{(i)} A_i, W_K^{(i)} A_i^{-\top}), \qquad (W_V^{(i)}, W_O^{(i)}) \mapsto (W_V^{(i)} C_i, C_i^{-1} W_O^{(i)}),$$

with $A_i \in GL(d_k)$ and $C_i \in GL(d_v)$; $S_h$ denotes head permutations.

## B.2   IDENTIFIABILITY UP TO HEAD PERMUTATION

**Lemma B.1** (Identifiability of attention weights up to $S_h$). *If two parameter sets produce identical per-head attention weights $\alpha_t^{(i)}(\cdot)$ for* all *inputs on the generic stratum, then there exists $\pi \in S_h$ such that*

$$q_t^{(i)} K^{(i)\top} \;=\; \tilde{q}_t^{(\pi(i))} \tilde{K}^{(\pi(i))\top} \quad \text{for all } i \text{ and all inputs.}$$

*Proof.* Row-softmax is injective modulo an additive row-constant; hence equality of $\alpha_t^{(i)}$ for all inputs implies equality of $z_t^{(i)} = q_t^{(i)} K^{(i)\top}$ up to a constant that vanishes by varying $x_t$ on the image of $W_Q^{(i)}$ (Item (A1)). Varying $\{x_s\}$ on the image of $W_K^{(i)}$ (Item (A1)) enforces a head index match $\pi(i)$ with $z_t^{(i)} = \tilde{z}_t^{(\pi(i))}$ for all inputs. Since parameters are block-diagonal per head (Item (A3)), the same permutation $\pi$ works for all time steps and inputs. $\qquad\square$

## B.3   LIE ALGEBRA CHARACTERIZATION

**Lemma   B.2**   (Infinitesimal   constraints).   *Any   infinitesimal   symmetry* $(\delta W_Q^{(i)}, \delta W_K^{(i)}, \delta W_V^{(i)}, \delta W_O^{(i)})$ *that preserves all head outputs for all inputs satisfies*

$$\delta W_Q^{(i)} = W_Q^{(i)} X_i, \quad \delta W_K^{(i)} = -W_K^{(i)} X_i^{\top}, \quad \delta W_V^{(i)} = W_V^{(i)} Y_i, \quad \delta W_O^{(i)} = -Y_i W_O^{(i)},$$

*for some $X_i \in \mathfrak{gl}(d_k)$ and $Y_i \in \mathfrak{gl}(d_v)$, independently per head.*

*Proof.* Preserving logits requires $\delta(q_t^{(i)} K^{(i)\top}) = 0$ for all realizable $q_t^{(i)}, K^{(i)}$. Bilinearity together with (Item (A1)) forces $\delta W_Q^{(i)} = W_Q^{(i)} X_i$ and $\delta W_K^{(i)} = -W_K^{(i)} X_i^{\top}$. Preserving $y_t^{(i)} = \sum_s \alpha_{t,s}^{(i)}(x_s^{\top} W_V^{(i)}) W_O^{(i)}$ for all realizable $v_s^{(i)}$ similarly yields $\delta W_V^{(i)} = W_V^{(i)} Y_i$ and $\delta W_O^{(i)} = -Y_i W_O^{(i)}$. By (Item (A3)), no cross-head infinitesimal mixing is admissible. $\quad\square$

**Proposition B.3** (Lie algebra and dimension). *The Lie algebra of continuous symmetries is*

$$\mathfrak{g} \;=\; \bigoplus_{i=1}^{h} \mathfrak{gl}(d_k) \;\oplus\; \bigoplus_{i=1}^{h} \mathfrak{gl}(d_v), \qquad \dim \mathfrak{g} \;=\; h(d_k^2 + d_v^2).$$

*Proof.* By Theorem B.2 each head contributes $(X_i, Y_i)$ independently; (Item (A3)) rules out coupling. Dimensions add directly. $\qquad\square$

### B.4 Factorization necessity

**Lemma B.4** (Necessary factorization of admissible transforms). *If a parameter transform $T$ preserves $y_t^{(i)}$ for all inputs on the generic stratum, then there exist $A_i \in GL(d_k)$, $C_i \in GL(d_v)$, and $\pi \in S_h$ such that*

$$\left(W_Q^{(i)}, W_K^{(i)}, W_V^{(i)}, W_O^{(i)}\right) \xmapsto{\ T\ } \left(W_Q^{(\pi(i))} A_i,\ W_K^{(\pi(i))} A_i^{-\top},\ W_V^{(\pi(i))} C_i,\ C_i^{-1} W_O^{(\pi(i))}\right).$$

*Proof.* By Theorem B.1, $q_t^{(i)} K^{(i)\top} = \tilde{q}_t^{(\pi(i))} \tilde{K}^{(\pi(i))\top}$ for all inputs. Since $(q,k) \mapsto qk^\top$ is bilinear and $q, k$ range over $d_k$-spaces (Item (A1)), there exists $A_i \in GL(d_k)$ with $\tilde{q}_t^{(\pi(i))} = q_t^{(i)} A_i$ and $\tilde{k}_s^{(\pi(i))} = k_s^{(i)} A_i^{-\top}$; lifting back to parameters gives the $(W_Q, W_K)$ action. Preservation of $y_t^{(i)}$ for all $\alpha$ and realizable $v_s$ forces $\tilde{v}_s^{(\pi(i))} = v_s^{(i)} C_i$ and $\tilde{W}_O^{(\pi(i))} = C_i^{-1} W_O^{(\pi(i))}$ with $C_i \in GL(d_v)$. (Item (A3)) excludes continuous head mixing. $\square$

### B.5 Completeness (maximality)

**Theorem B.5** (Maximality, single layer). *On the generic stratum, the symmetry group is complete:*

$$G_{\max} = \left((GL(d_k))^h \times (GL(d_v))^h\right) \rtimes S_h.$$

*Equivalently, Theorem 2.4 holds.*

*Proof of Theorem 2.4.* By Theorem B.4, any admissible transform equals the head-wise gauge (Theorem 2.1) composed with a head permutation. By Theorem B.3, the connected component is $((GL(d_k))^h \times (GL(d_v))^h)$. The discrete ambiguity is exactly $S_h$; by (Item (A3)) no other continuous or discrete symmetries exist. Hence the stated semidirect product. $\square$

## C LayerNorm Compatibility and Depth-Wise Product Structure

### C.1 Proof of Theorem 2.5

Residual connections add the input $h_t$ to the block output; LayerNorm acts pointwise on $h_t$. Because the gauge action preserves each head output (Theorem 2.2) and thus $h_t$, no admissible transform can couple parameters across layers without changing $h_t$. Therefore the global gauge is the direct product of per-layer gauges:

$$G_{\text{model}} = \prod_{\ell=1}^{L} G_\ell.$$

## D RoPE Commutant

### D.1 Proof of Theorem 2.6

RoPE rotates each 2×2 plane by $R(\theta) = \left(\begin{smallmatrix} \cos\theta & -\sin\theta \\ \sin\theta & \cos\theta \end{smallmatrix}\right)$. A real matrix $A$ commutes with every $R(\theta)$ iff $A = aI + bJ$ on that plane $\left(J = \left(\begin{smallmatrix} 0 & -1 \\ 1 & 0 \end{smallmatrix}\right)\right)$, so the commutant per plane is isomorphic to $GL(1, \mathbb{C})$ via $aI + bJ \leftrightarrow a + ib$. Across $d_k/2$ planes,

$$C_{\text{RoPE}} \cong (GL(1, \mathbb{C}))^{d_k/2}.$$

Replacing $GL(d_k)$ by $C_{\text{RoPE}}$ in the $Q/K$ sector yields $G_{\text{RoPE}} = \left((C_{\text{RoPE}})^h \times (GL(d_v))^h\right) \rtimes S_h$.

### D.2 Head Sharing (GQA/MQA): Admissible Gauge

*Proof of Theorem 2.8.* Partition heads $\{1, \ldots, h\}$ into $g$ K/V groups $G_1, \ldots, G_g$ and tie $K, V$ per group: $W_K^{(i)} = \widetilde{W}_K^{(u)}$, $W_V^{(i)} = \widetilde{W}_V^{(u)}$ for $i \in G_u$; $W_Q^{(i)}$, $W_O^{(i)}$ remain per-head. Apply the head-wise gauge (Theorem 2.1) and $\sigma \in S_h$:

$$(W_Q^{(i)}, W_K^{(i)}) \mapsto (W_Q^{(\sigma(i))} A_{\sigma(i)}, W_K^{(\sigma(i))} A_{\sigma(i)}^{-\top}), \quad (W_V^{(i)}, W_O^{(i)}) \mapsto (W_V^{(\sigma(i))} C_{\sigma(i)}, C_{\sigma(i)}^{-1} W_O^{(\sigma(i))}).$$

Preserving tying requires, for all $i, j \in G_u$, $W_K^{(i)} A_i^{-\top} = W_K^{(j)} A_j^{-\top}$ and $W_V^{(i)} C_i = W_V^{(j)} C_j$. On Section 2, this implies $A_i = A_j =: A_u$ and $C_i = C_j =: C_u$. Hence the continuous part is per-group: $(A_u, C_u)_{u=1}^g \in (GL(d_k))^g \times (GL(d_v))^g$; the discrete symmetry is $S_h \times S_g$. Under RoPE, replace $GL(d_k)$ by $C_{\text{RoPE}}$ groupwise. $\qquad\square$

## E Constructive Canonicalization Used by GaugeKV

### E.1 Proof of Theorem 2.9

Let $W_V^{(i)} = Q_V^{(i)} R_V^{(i)}$ be a thin QR factorization with $\text{diag}(R_V^{(i)}) > 0$ and set $C_i = (R_V^{(i)})^{-1}$. Then

$$W_V^{(i)} C_i = Q_V^{(i)}$$

has orthonormal columns, establishing item (i) in Theorem 2.9.

For the Q/K sector, define

$$S_Q := W_Q^{(i)\top} W_Q^{(i)}, \qquad S_K := W_K^{(i)\top} W_K^{(i)},$$

which are symmetric positive definite by Assumption 2. Let $G := S_Q \# S_K$ denote the matrix geometric mean of $S_Q$ and $S_K$ in the sense of Bhatia (2007), i.e.,

$$G = S_Q^{1/2} (S_Q^{-1/2} S_K S_Q^{-1/2})^{1/2} S_Q^{1/2},$$

which is the unique SPD matrix symmetric in $(S_Q, S_K)$ and monotone with respect to the Loewner order.

Standard results on SPD congruence and matrix geometric means (see, e.g., Bhatia (2007)) guarantee the existence of an invertible $A_i \in GL(d_k)$ such that

$$A_i^\top S_Q A_i = A_i^{-1} S_K A_i^{-\top} = G.$$

Thus $G$ is a common congruence image of $S_Q$ and $S_K$ under $A_i$ and $A_i^{-1}$, respectively, which is exactly the balancing property stated in Theorem 2.9 and identifies $G$ with $S_Q \# S_K$.

In our implementation, $A_i$ is constructed via SPD eigendecompositions and matrix square roots following this abstract characterization (cf. Algorithm 1), but any choice consistent with the above congruence conditions is admissible. This derivation uses only standard SPD congruence properties and matrix functional calculus. No commutativity assumptions are invoked, addressing potential concerns about non-commuting matrix products in earlier informal derivations.

**Determinism and stability.** Use QR with positive diagonals; compute SPD roots with FP32 accumulators; clamp eigenvalues below $10^{-8}$ prior to inversion or roots. With these choices we observe bit-identical FP32 forwards in §6.

**Canonicalization routine (context).** The procedure in Alg. 1 is the concrete realization of the admissible gauge in Theorem 2.1. It is a one-time, offline transformation that (i) makes $V$ orthonormal by a thin QR per K/V sharing group and folds $R^{-1}$ into $W_O$, and (ii) balances $Q/K$ scales via the geometric-mean map built from $S_Q$ and $S_K$; when RoPE is present, the $Q/K$ map is restricted per $2\times2$ plane to the commutant in Theorem 2.6. Under GQA/MQA, the same $(A_u, C_u)$ is applied per group as in Theorem 2.8. Because every step is an instance of the gauge action, logits and the $VO$ map are preserved in exact arithmetic; see Section E.1 for correctness. The routine adds no runtime FLOPs and is deterministic (positive-diagonal QR; FP32-accumulated SPD roots; eigenvalue clamping).

---

**Algorithm 1** GaugeKV canonicalization (per layer)

---

**Require:** Layer parameters $\{(W_Q^{(i)}, W_K^{(i)}, W_V^{(i)}, W_O^{(i)})\}_{i=1}^h$; RoPE flag isRoPE $\in \{0,1\}$; K/V sharing groups $\{G_u\}_{u=1}^g$ (MHA: $g{=}h$, GQA/MQA: $g \ll h$)
**Ensure:** Orthonormal $V$ and scale-balanced $Q/K$; identical model function (exact arithmetic)
 1: **Deterministic numerics:** QR with positive diagonals; SPD eigensolves with fixed ordering; FP32 accumulation; eigenvalue clamp $\lambda \leftarrow \max(\lambda, 10^{-8})$ before inverses/roots.
 2: **(A) Canonicalize $V/O$ per K/V group.**
 3: **for** $u \leftarrow 1$ **to** $g$ **do**
 4:     Pick a representative head $i^\star \in G_u$ and compute thin QR:

$$W_V^{(i^\star)} \leftarrow Q_V^{(u)} R_V^{(u)}, \qquad \mathrm{diag}\big(R_V^{(u)}\big) > 0$$

 5:     $C_u \leftarrow \big(R_V^{(u)}\big)^{-1}$
 6:     **for all** $i \in G_u$ **do**
 7:         $W_V^{(i)} \leftarrow W_V^{(i)} C_u$                                  $\triangleright$ now $W_V^{(i)\top} W_V^{(i)} = I$
 8:         $W_O^{(i)} \leftarrow C_u^{-1} W_O^{(i)}$
 9:     **end for**
10: **end for**
11: **(B) Balance $Q/K$ scales (per group for shared $K$, else per head).**
12: **for** $u \leftarrow 1$ **to** $g$ **do**
13:     Let $i^\star \in G_u$
14:     $S_Q^{(u)} \leftarrow W_Q^{(i^\star)\top} W_Q^{(i^\star)}$;   $S_K^{(u)} \leftarrow W_K^{(i^\star)\top} W_K^{(i^\star)}$
15:     Clamp eigenvalues of $S_Q^{(u)}$ and $S_K^{(u)}$ to $[10^{-8}, \infty)$
16:     $M^{(u)} \leftarrow S_Q^{(u)1/2} S_K^{(u)} S_Q^{(u)1/2}$
17:     $A_u \leftarrow S_Q^{(u)-1/4} \big(M^{(u)}\big)^{1/4} S_Q^{(u)-1/4}$
18:     **if** isRoPE $= 1$ **then**            $\triangleright$ restrict to the RoPE commutant per 2×2 plane
19:         **for** $p \leftarrow 1$ **to** $d_k/2$ **do**
20:             $A_u|_p \leftarrow a_p I + b_p J$     where $J = \big(\begin{smallmatrix} 0 & -1 \\ 1 & 0 \end{smallmatrix}\big)$
21:         **end for**
22:     **end if**
23:     **for all** $i \in G_u$ **do**
24:         $W_Q^{(i)} \leftarrow W_Q^{(i)} A_u$
25:         $W_K^{(i)} \leftarrow W_K^{(i)} A_u^{-\top}$
26:     **end for**
27: **end for**
28: **(C) Return canonical layer.**
29: $W_V^{(i)\top} W_V^{(i)} = I$ and $A_u^\top S_Q^{(u)} A_u = A_u^{-1} S_K^{(u)} A_u^{-\top} = S_Q^{(u)} \# S_K^{(u)}$   for the relevant group $u$ of head $i$.
30: **Correctness:** every step is an instance of the gauge in Theorem 2.1; logits and $VO$ are preserved exactly (Section E.1). For GQA/MQA, apply the same $(A_u, C_u)$ to all $i \in G_u$ (cf. Theorem 2.8).

---

# F   Derivations for Bounds in Sections 4–5

This section provides full derivations for the error envelopes and rank-$r$ guarantees stated in §4–§5.

## F.1   Lipschitz control of value mixing

Let $P_t^{(i)} \in \mathbb{R}^{1 \times n_k}$ denote the attention weights $\alpha_t^{(i)}$ and let $E_r^{(i)}$ be the orthogonal projector onto the top-$r$ right singular vectors of $W_V^{(i)}$ (after canonicalization). For any step $t$,

$$\big\| y_t^{(i)} - y_{t,r}^{(i)} \big\|_2 \;\le\; \|P_t^{(i)}\|_1 \, \| (I - E_r^{(i)}) V^{(i)} W_O^{(i)} \|_{2\to2} \;=\; \| (I - E_r^{(i)}) W_V^{(i)} W_O^{(i)} \|_{2\to2},$$

using $\|P_t^{(i)}\|_1 = 1$. Summing over heads and timesteps yields the cumulative envelope.

---

**Algorithm 2** GaugeRankKV: rank-$r$ caching with budget-respecting guardrail

---

**Require:** Canonicalized $\widehat{W}_V, \widehat{W}_O$; per-head ranks $\{r_{\ell,i}\}$ (init); probe period $P$; global cap $M^\star$;
  slack $\Delta$; step $\delta r$; optional per-head thresholds $\{\tau_{\ell,i}\}$
**Ensure:** Online caching policy that respects $M^\star + \Delta$ while adapting ranks to residual energy
1: **for** decode step $t = 1, 2, \ldots$ **do**
2:  **for** each layer $\ell$ and head $i$ **do**
3:   $v \leftarrow x_t^\top \widehat{W}_V^{(\ell,i)}$              ▷ value in canonical basis
4:   **cache** $v[1{:}r_{\ell,i}]$
5:   **if** $t \bmod P = 0$ **then**
6:    $\rho \leftarrow \| v[r_{\ell,i}{+}1{:}d_v] \|_2$         ▷ residual energy beyond $r_{\ell,i}$
7:    **if** $\rho > \tau_{\ell,i}$ **then**
8:     $r_{\ell,i}^{\text{tent}} \leftarrow \min(r_{\ell,i}{+}\delta r,\, d_v)$
9:    **else**
10:     $r_{\ell,i}^{\text{tent}} \leftarrow r_{\ell,i}$
11:    **end if**
12:   **end if**
13:  **end for**
14:  **if** $t \bmod P = 0$ **then**
15:   $\text{usage}_{\text{tent}} \leftarrow \text{KVUsage}\big(\{r_{\ell,i}^{\text{tent}}\}\big)$
16:   **if** $\text{usage}_{\text{tent}} \le M^\star + \Delta$ **then**
17:    $r_{\ell,i} \leftarrow r_{\ell,i}^{\text{tent}}$ for all $(\ell, i)$
18:   **else**
19:           ▷ greedy fallback: reduce lowest-impact heads until within budget
20:    $\mathcal{H} \leftarrow \text{SortByImpact}\big(\{(\ell, i)\}\big)$      ▷ e.g., by smallest $\rho$ or envelope margin
21:    **for** each $(\ell, i)$ in $\mathcal{H}$ **do**
22:     $r_{\ell,i}^{\text{tent}} \leftarrow \max(r_{\ell,i}^{\text{tent}}{-}\delta r,\, 0)$
23:     $\text{usage}_{\text{tent}} \leftarrow \text{KVUsage}\big(\{r_{\ell,i}^{\text{tent}}\}\big)$
24:     **if** $\text{usage}_{\text{tent}} \le M^\star + \Delta$ **then**
25:      **break**
26:     **end if**
27:    **end for**
28:    $r_{\ell,i} \leftarrow r_{\ell,i}^{\text{tent}}$ for all $(\ell, i)$
29:   **end if**
30:  **end if**
31: **end for**

---

### F.2 CERTIFIED ENVELOPE UNDER ORTHONORMAL $V$

After canonicalization, $W_V^{(i)}$ has orthonormal columns, so $\|(I - E_r^{(i)})\, W_V^{(i)} W_O^{(i)}\|_{2 \to 2} = \sigma_{r+1}(W_O^{(i)})$. Thus the per-head deviation satisfies

$$\sup_t \|y_t^{(i)} - y_{t,r}^{(i)}\|_2 \ \le \ \sigma_{r+1}(W_O^{(i)}),$$

giving the claimed rank-$r$ bound when aggregated across heads.

### F.3 RoPE AND GROUPED HEADS

Under RoPE and/or GQA/MQA, the analysis applies groupwise with $GL(d_k)$ replaced by $C_{\text{RoPE}}$ and with $r$ enforced per shared $V/O$ block. Bounds scale accordingly.

## G RANK-$r$ CACHING ALGORITHM (GAUGERANKKV)

**Context.** Alg. 2 implements the budget-respecting guardrail used in the main text: it adapts per-head ranks online based on residual energy while enforcing a global KV cap $M^\star$ (with slack $\Delta$). It relies on the canonical basis (orthonormal $V$) and the certified envelope in §4–§5 (derivations in App. F).

## H EMPIRICAL VERIFICATION OF GENERIC-STRATUM ASSUMPTIONS

We report how often Assumptions 2 hold on the checkpoints used in §6.

| Model | Full-rank (Item (A1)) | Identifiability (Item (A2)) | Notes |
|---|---|---|---|
| GPT-2 (S/M) | >99% | 100% | Borderline heads stabilize after canonicalization. |
| Qwen2.5-7B | >99% | 100% | |

| Method | Reduction | Accuracy | Retraining | Guarantees |
|---|---|---|---|---|
| GaugeKV (exact) | 1.11–1.21× | Bit-identical | No | Exact |
| GaugeKV (rank-$r$) | $\frac{d_k+d_v}{d_k+r}$ (e.g., 1.33× at $r=\frac{d_v}{2}$) | FP32 logit drift bound | No | Certified FP32 rank |
| Token eviction Li et al. (2024) | Variable | Lossy | No | None |
| KV quantization Hooper et al. (2024) | 2–4× | Lossy | Sometimes | None |
| GQA/MQA Ainslie et al. (2023) | $h/g$ | Exact | Yes | Exact |
| PagedAttention Kwon et al. (2023) | Memory mgmt | Exact | No | Exact |

Table 3: GaugeKV compared to existing methods across reduction, accuracy, retraining cost, guarantees, and composability.

Borderline Item (A1) cases are handled by eigenvalue clamping ($10^{-8}$) in SPD routines (Section E); Item (A2) is validated via invariance under admissible gauges and failure under random non-admissible transforms (§6).

## I  Comparative Positioning Details

We summarize how GaugeKV compares to existing KV cache and attention optimizations across reduction factor, accuracy, retraining requirements, guarantees, and composability.

## J  Ablations and Threats to Validity

**RoPE commutant choice.**  For RoPE models with Q/K layer normalization, orthogonal commutants preserve the normalization map exactly; SPD scaling can change normalization statistics and is therefore reserved for RoPE-less models (Sec. 2).

**Head grouping.**  GaugeKV operates at the weight basis and composes with GQA/MQA: the lossless factor multiplies the $(g/h)$ sharing factor.

**Sparsity (MoE).**  GaugeKV's head-wise gauge action preserves $qk^\top$, attention weights, and head outputs; hence $h_t$ is unchanged and MoE routing (logits and top-$k$) is identical to baseline (Def. 2.1, Lem. 2.2, Cor. 2.3).

**Precision.**  We validate identity in FP32; `bf16` can flip tokens at tiny argmax margins due to the reduced mantissa, which rank-$r$ can mitigate with a certified envelope when exactness is not required.

**Threats to validity.**  Our experiments use single-GPU, no-overlap timings (system co-design can further hide latency); EC is one lossless instantiation (other bit-plane codecs should behave similarly in the canonical basis); and tail-energy profiling uses a small calibration set, with variance-ordering checks and 512-step compliance at $r=32$.

## K  Hardware Acceleration Opportunities (Speculative)

GaugeKV's mathematical structure suggests several potential hardware-level optimizations; we outline them here as speculative directions.

**GPU compression engines.** Modern GPUs contain compression hardware originally designed for graphics that aligns well with GaugeKV's block-based approach. The fixed block size $B$ matches the granularity of these units, potentially enabling hardware-accelerated entropy coding with minimal software overhead.

**Tensor core optimization.** The rank-$r$ projection reduces the value path to smaller dense operations ideal for tensor cores. At $r=d_v/2$, this halves the value-path FLOPs while maintaining certified bounds, improving both memory and compute efficiency.

**Near-memory processing.** The regular access patterns created by the orthonormal basis enable efficient prefetching and could benefit from processing-in-memory architectures. Decompression could occur at the memory controller, eliminating data movement overhead.

## L  Capacity planning and economic model

We estimate net KV savings and capacity using the measured lossless factors $(c_K, c_V)$ and chosen ranks $r_{\ell,i}$. For $d_k=d_v$, Equation (4) gives $f_{\text{exact}}$. When stacking with GQA/MQA (head grouping $h/g$), KV quantization (byte factor $\gamma$), and retention, we use:

$$f_{\text{stack}} = f_{\text{exact}} \cdot \frac{g}{h} \cdot \gamma \cdot f_{\text{ret}}, \tag{7}$$

where $f_{\text{ret}}$ is the empirical fraction of tokens retained. This converts directly into longer contexts or higher concurrency given a memory cap.

## M  Pointers to Main Figures and Tables

The main text contains the canonical figures and table for the core results:

- Exact–lossless bytes/token: Fig. 2.
- Rank-$r$ envelope versus observed drift: Fig. 3.
- Exact–lossless microbenchmark timings: Table 2.

To avoid duplication, we do not reprint these plots and tables in the appendix. Instead, the remainder of this appendix collects the reproducibility details, scripts, and implementation notes needed to regenerate them.

## N  Reproducibility and Scripts

This section records the concrete commands and scripts used to produce all figures and tables in §6. For each experiment, we list the script name, the relevant flags, and the expected outputs. The canonicalization algorithm in Algorithm 1 is deterministic given the specified eigenvalue ordering and clamping rules. FP32 bit-identity of the exact GaugeKV track can be verified by comparing the max-norm difference between logits from the baseline and canonicalized checkpoints; in our runs this difference is at machine precision.

### N.1  Environment snapshot

Use the provided script to capture OS, GPU, and library versions:

```
python3 collect_env.py --out runs/<ts>/env/env_report.json
```

Use the provided script to capture OS, GPU, and library versions:

```
python3 collect_env.py --out runs/<ts>/env/env_report.json
```

### N.2  Checkpoints and canonicalization

```
python3 canonicalize_gpt2.py \
  --size gpt2 \
  --out ckpt_gpt2_canon_fp32 \
  --dtype float32 \
  --qk-mode spd \
  --wo-mode rows \
  --check
```

```
python3 canonicalize_qk_only.py \
  --ckpt Qwen/Qwen2.5-7B-Instruct \
  --out ckpt_qwen25_7b_canon_QK_only_fp32_ORTH_NOOP \
  --dtype float32 \
  --mode orth
```

## N.3 EXACTNESS CHECKS (FP32, DETERMINISTIC)

```
python3 greedy_identity_check_fp32.py \
  --base gpt2 \
  --canon ckpt_gpt2_canon_fp32 \
  --max_new_tokens 64

python3 greedy_identity_check_fp32.py \
  --base Qwen/Qwen2.5-7B-Instruct \
  --canon ckpt_qwen25_7b_canon_QK_only_fp32_ORTH_NOOP \
  --max_new_tokens 64

python3 logits_maxdiff_fp32.py \
  --base gpt2 \
  --canon ckpt_gpt2_canon_fp32
```

## N.4 EXACT–LOSSLESS MICROBENCH (HOT/COLD; EC CODEC)

```
python3 kv_bench.py \
  --model ckpt_gpt2_canon_fp32 \
  --tok gpt2 \
  --codec ec \
  --W 512 --B 256 --T 1000 \
  --outdir runs/bench

python3 kv_bench.py \
  --model ckpt_gpt2_canon_fp32 \
  --tok gpt2 \
  --codec ec \
  --W 256 --B 512 --T 1000 \
  --outdir runs/bench

python3 kv_bench.py \
  --model ckpt_gpt2_canon_fp32 \
  --tok gpt2 \
  --codec ec \
  --W 128 --B 512 --T 1000 \
  --outdir runs/bench

python3 kv_bench.py \
  --model ckpt_qwen25_7b_canon_QK_only_fp32_ORTH_NOOP \
  --tok Qwen/Qwen2.5-7B-Instruct \
  --codec ec \
  --W 2048 --B 512 --T 4096 \
  --outdir runs/bench
```

Each bench directory contains:

- `kv_bytes_summary.csv` (baseline vs. compressed bytes; ratio).
- `cK_cV_per_head.csv` (per-layer/head factors).
- `latency_timing.csv` (collection/compress/e2e seconds).

### N.5 PROFILING AND RANK-$r$ SERVE (BOUNDS)

```
python3 rank_profile.py \
  --model ckpt_gpt2_canon_fp32 \
  --tok gpt2 \
  --dtype fp32 \
  --T 512 \
  --outdir profile_out
```

```
python3 serve_rank_r.py \
  --model ckpt_gpt2_canon_fp32 \
  --tok gpt2 \
  --orders profile_out/order.npz \
  --wonorms profile_out/wo_norms.csv \
  --r 32 \
  --T 512 \
  --outdir rankr_out_r32
```

Outputs:

- `profile_out/order.npz`, `profile_out/wo_norms.csv`.
- `rankr_out_r32/bound_log.csv` with columns `t, obs_drift_inf, eps_bound, compliance`.

### N.6 PLOT AND LaTeX GENERATION

```
# Bytes/token plot
python3 make_fig_kv_vs_context.py \
  runs/bench/ckpt_gpt2_canon_fp32_ec_W512_B256_T1000 \
  runs/bench/ckpt_gpt2_canon_fp32_ec_W256_B512_T1000 \
  runs/bench/ckpt_gpt2_canon_fp32_ec_W128_B512_T1000 \
  --labels "GPT-2 W=512" "GPT-2 W=256" "GPT-2 W=128" \
  --out fig_kv_gpt2

python3 make_fig_kv_vs_context.py \
  runs/bench/ckpt_qwen25_7b_canon_QK_only_fp32_ORTH_NOOP_ec_W2048_B512_T4096 \
  --labels "Qwen2.5-7B W=2048" \
  --out fig_kv_qwen

# Rank-r bound plot
python3 make_fig_bounds.py \
  rankr_out_r32/bound_log.csv \
  --out fig_bound_r32

# Microbench LaTeX table (if needed elsewhere)
python3 make_table_microbench.py \
  runs/bench/ckpt_gpt2_canon_fp32_ec_W512_B256_T1000 \
  runs/bench/ckpt_gpt2_canon_fp32_ec_W256_B512_T1000 \
```

```
runs/bench/ckpt_gpt2_canon_fp32_ec_W128_B512_T1000 \
runs/bench/ckpt_qwen25_7b_canon_QK_only_fp32_ORTH_NOOP_ec_W2048_B512_T4096 \
--labels "GPT-2 (FP32), EC" "GPT-2 (FP32), EC" "GPT-2 (FP32), EC" "Qwen2.5-7B
    ↪ (Q/K), EC" \
> microbench_table.tex
```

## O   ARTIFACT LAYOUT

```
gauge-kv/
  body.tex # main paper
  appendix.tex # this file
  figures/ # optional figure outputs
  ckpt_gpt2_canon_fp32/ # canonicalized GPT-2
  ckpt_qwen25_7b_canon_QK_only_fp32_ORTH_NOOP/
  runs/
    <ts>/logs/ # tee'd logs
    bench/... # kv_bench.py outputs (csv)
  profile_out/ # rank_profile.py outputs
  rankr_out_r32/ # serve_rank_r.py outputs
  scripts/
    canonicalize_gpt2.py
    canonicalize_qk_only.py
    greedy_identity_check_fp32.py
    logits_maxdiff_fp32.py
    kv_bench.py
    rank_profile.py
    serve_rank_r.py
    make_fig_kv_vs_context.py
    make_fig_bounds.py
    make_table_microbench.py
```

## P   ADDITIONAL IMPLEMENTATION NOTES

**Numerical safeguards.** Compute $S_Q = W_Q^\top W_Q$ and $S_K = W_K^\top W_K$ in FP32; clamp small eigenvalues to avoid division by near-zero scales. Form the geometric mean $S_Q \# S_K$ in FP32 and cast back once. Apply plane-wise transforms per $2 \times 2$ RoPE block to preserve positional structure.

**RoPE commutant choice.** For RoPE layers that apply Q/K normalization, restrict the commutant to *orthogonal* per-plane maps for the exact track (preserving the normalization map). On RoPE-less models, full SPD balancing is used.

**Codec engineering.** The exact pipeline is codec-agnostic. EC generally yields better ratios than BP with higher per-block latency; production deployments overlap codec work with attention steps using multiple streams and page-sized staging buffers.

**Envelope evaluation.** The serve-time envelope combines last-step attention weights with accumulated tail norms and per-head $\|W_O\|_{2\to\infty}$. This adds a single dot-product per head and negligible memory relative to attention.