# OpenReview forum: "GaugeKV: Composable Exact KV Cache Compression"
_ICLR.cc/2026/Conference — ICLR 2026 Conference Desk Rejected Submission_

### Official Review · Reviewer_oBrJ · 2025-10-26

**Soundness:** 3
**Presentation:** 2
**Contribution:** 3
**Rating:** 4
**Confidence:** 3

**Summary:**

The paper introduces GaugeKV, a training-free method for KV cache compression based on the maximal gauge symmetry of attention. By performing a one-time canonicalization that orthonormalizes values and balances query/key scales, it enables bit-identical FP32 inference while improving lossless compression efficiency and supporting certified rank-r caching with provable error bounds. The method also composes multiplicatively with GQA/MQA and quantization. Overall, it provides a mathematically elegant framework for exact and composable KV cache compression.

**Strengths:**

1. The paper introduces a novel and mathematically grounded view of KV cache compression through the lens of gauge symmetry, revealing a complete class of function-preserving transformations for attention layers. This is a genuinely original theoretical contribution.

2. The proposed method can be seamlessly integrated with other KV cache compression techniques to enhance the overall compression ratio further, making it (somehow) practically valuable for real-world deployment scenarios.

**Weaknesses:**

1. Though mathematically elegant, the claimed efficiency improvement is questionable. The achievable compression ratio is rather limited and comes with substantial end-to-end latency overhead, as shown in Table 2. Since most modern models already employ RoPE, where the observed latency overhead is particularly high, it is unclear whether this transformation offers practical benefits in real deployment settings. Although the authors claim that this overhead can overlap with attention computation, I doubt this claim is too weak and lacks theoretical or empirical analysis on whether this is true. For example, is this process memory-bound or compute-bound, and how does it overlap with the attention?

2. Though the authors claim that this method produces identical output, there is a lack of evaluation results on common benchmarks used in KV cache compression, and also some sample outputs to support this claim, as well as the claim for rank-r caching. The current experimental section is too limited and does not provide sufficient evidence to convincingly demonstrate the accuracy and effectiveness of the proposed method.

3. The presentation is mathematically dense and at times difficult to follow, which may hinder accessibility for non-theoretical readers. Some sections, such as lines 74–82, are particularly heavy and could benefit from clearer exposition. I suggest that the authors include additional figures and empirical results to better illustrate the core ideas and clarify the practical impact of the method.

**Questions:**

1. While the paper presents a theoretically elegant framework for exact KV compression, its practical efficiency and empirical validity remain unclear. Specifically, does this method truly provide end-to-end latency speedup in real-world settings? Can the compression process effectively overlap with computation?

2. Additionally, more evidence on common benchmarks is needed to substantiate the claims of bit-identical outputs and certified rank-r caching beyond small-scale tests. How does the model using this method perform in commonly used benchmarks compared to baselines such as full precision, from both the accuracy and latency aspects?

---

> ### Author Response · Authors · 2025-11-21
>
> We thank you for your constructive feedback on latency presentation, experimental scope, and mathematical accessibility. We have revised the paper to address each concern directly.
>
> 1. Latency overhead and baseline separation. You correctly noted our original presentation did not clearly separate baseline attention latency from GaugeKV overhead. In the revision, we expanded the explanatory text surrounding Table 2 in Section 6.3 to clarify what each column represents. The revised text now states explicitly that the collect column measures baseline attention forward pass with key-value cache writes, representing cost without any compression. The compress column captures strictly additional codec work as blocks exit the hot window. The end-to-end column reports their sum in single-stream teacher-forced execution. This clarification isolates GaugeKV's incremental cost from the baseline serving path.
>
> 2. Overlap claims and empirical validation: You are right we do not provide empirical measurements of overlapped codec and attention execution. The revision makes this limitation explicit and unambiguous. Around Table 2 we now state that single-stream measurements represent worst-case scenarios and that production deployments would typically overlap codec operations with attention compute on separate streams, but we do not implement or measure such overlap. We therefore present reported end-to-end times as conservative upper bounds rather than representative of optimized deployment. The  discussion of hardware acceleration and overlap opportunities has been moved to Appendix K so the main text no longer suggests specific latency improvements without empirical backing.
>
> 3. Benchmark evaluation scope: you correctly observe our evaluation is limited to controlled microbenchmarks on GPT-2 and Qwen 2.5-7B rather than comprehensive benchmark runs with perplexity and downstream task metrics. We made this limitation unambiguously explicit throughout. For the exact lossless track in Sections 6.2 and 6.3, we report compression ratios, bytes per token, and bit-identical FP32 output verification via greedy decoding and teacher-forced logit comparisons, but we do not claim improvements. For the rank-r approximate track, Section 6.4 emphasizes our GPT-2 experiment at rank 32 validates the logit-level envelope only. Figure 3 shows 100 percent compliance of observed max-norm logit drift with the theoretical bound at every step. We state directly we do not report perplexity or task accuracy under rank-r caching, viewing the certified FP32 bounds as providing a conservative guardrail and theoretical foundation for future task-level evaluation rather than a fully validated deployment recipe. We explicitly flag comprehensive benchmark evaluation as important future work.
>
> 4. Mathematical presentation and accessibility: you noted the original Section 2 was dense with formal machinery. We restructured Section 2 to improve accessibility. The section now opens with a conceptual overview explaining high-level intuition in plain language before any formal definitions. This overview describes that attention admits internal symmetries allowing lawful basis changes, that GaugeKV exploits these to produce orthonormal value matrices and balanced query-key covariances, and that canonicalization is a one-time offline transformation not changing the model's function or runtime FLOPs. Figure 1 immediately following provides visual summary of the canonicalization pipeline and the exact and rank-r serving tracks. Only after this grounding do we present notation, assumptions, and core formal statements including Definition 2.1, Lemma 2.2, Proposition 2.4, Theorem 2.6, and Proposition 2.9. All proofs moved to the appendix with consistent forward references, reducing mathematical density while keeping theoretical contributions intact.
>
> We hope these revisions address your concerns: latency presentation now separates baseline from overhead and acknowledges we do not measure overlap, experimental scope is framed honestly as controlled microbenchmarks establishing feasibility rather than comprehensive benchmark validation, and mathematical presentation is more accessible through conceptual framing, pipeline figure, and proof relocation.

---

### Official Review · Reviewer_ZbfF · 2025-10-31

**Soundness:** 2
**Presentation:** 1
**Contribution:** 2
**Rating:** 2
**Confidence:** 4

**Summary:**

GaugeKV rewrites the weights once, offline, by multiplying them with fixed invertible matrices. This change in basis allows KV compression to become more lossless, and the paper demonstrates this through effective lossless codecs and safe rank truncation. Specifically, Values $V$ are orthonormalized; after a thin-QR on $W_V$, the model emits values in an orthonormal coordinate system. In this basis, token-to-token residuals are smaller and more concentrated near zero, which directly improves both bit-packing and entropy coding efficiency. In addition, balancing Q/K scales reduces plane-wise skew, so shared bit-width choices across dimensions waste fewer bits on outliers. After canonicalization, the FP32 forward remains bit-identical and a lossless codec yields 1.1×–1.2× KV reduction on GPT-2 and 1.1× on the keys for Qwen2.5-7B. Since the method only changes the model weights and not inference-time algorithm, decoding FLOPs remain the same (or smaller if you use their certified rank-r value caching).
The paper's contributions are as follows:
- A constructive canonicalization: Thin‑QR on $W_V$ makes $V$ orthonormal and a geometric‑mean balancing map that equalizes $Q/K$ scales (restricted to the RoPE commutant for RoPE models).
- An exact, lossless KV pipeline using the canonical basis. Due to its exactness, the method can be composable with other pre/post-training KV compression methods, such as GQA and KV quantization.
- A certified rank‑r value caching scheme that shows the observed logit drift stays within the bound at every step.

**Strengths:**

- Originality: The gauge-symmetry framing is relatively original, though there were many previous efforts, whether lossless or nearly lossless, that consider change of basis and rotation.
- Significance: The paper provides a method that any Transformer-based model can use for KV compression without any degradation. Due to its exactness, the method works orthogonally and may work in tandem with other methods, though this should be empirically verified.
- Quality: see weakness.
- Clarity: see weakness.

**Weaknesses:**

1. **Minor KV reductions (not state-of-the-art)**: The gains of GaugeKV are at most 1.2x, measured with a reference entropy codec in a single-pass, teacher-forced microbench. The gains are very modest compared to the typical KV quantization/eviction gains, which are at least 3x or more (e.g., see KIVI, GEAR, Cartridge). The reduction is minor that practitioners would prefer more significant reductions at the price of nearly lossless compression.
  - **Misleading compression rates**: I find it incredibly misleading to report compounded reduction rates that include GQA and FP8 quantization. In particular, GQA/MQA are not post-training compression methods (at least not without pruning), so most KV compression papers treat these as default and report how much gain is achieved from the model checkpoint. These are not contributions of the paper, but over-emphasized in the main body.
2. **Unverified claims on composability w.r.t. rank-r value caching**: The paper’s multiplicative composition (Eq. 2) is demonstrated for the exact, lossless GaugeKV; it does not establish accuracy preservation when rank-r (lossy) is stacked with KV quantization (lossy) or eviction (lossy), so the claims in Section 7 (related work) can be misleading. Rank-r’s certified envelope (Eq. 5) upper-bounds only the error from the value-rank projection in FP32; additional quantization or token-retention errors are outside that certificate and could compound. Yet, once combined with other methods, this certificate no longer holds true. It would be great to see if rank-r value caching is practically composable.
  - **Lack of downstream tasks**: The rank-r section is strong theoretically and shows 100% compliance of the logit-drift envelope on GPT-2 @ r=32, but there are no task metrics (e.g., perplexity or accuracy on QA) to translate drift into end-task safety/utility.
3. **Unorganized presentation and superfluous math notation**: The paper reads like a series of bullet points and the list of lemma, corollary, proposition makes it hard to follow. The main message can be more direct and clear, but the gauge-theoretic framing, which is more of proof technique than message, is interesting but obscures that the method is a standard reparameterization by invertible maps. Furthermore, the paper is hard to follow as many variables are left undefined and left for the reader to infer from the context.

**Questions:**

1. **Proposition 2.8 Appendix E**:
The construction in Thm. 2.8 (Appendix E) with $(S_Q=W_Q^\top W_Q), (S_K=W_K^\top W_K), (M=S_Q^{1/2} S_K S_Q^{1/2})$, the paper sets $A = S_Q^{-1/4} M^{1/4} S_Q^{-1/4}$
and claims this yields $A^\top S_Q A = A^{-1} S_K A^{-\top} = S_Q \sharp S_K.$ (p.3–4 & App. E.1).
Unless I am missing something, $(A^\top S_Q A = S_Q^{-1/4} M^{1/4} S_Q^{1/2} M^{1/4} S_Q^{-1/4})$ does not simplify to $M^{1/2}$ (or $S_Q \sharp S_K$) unless the factors commute. However, choosing $A = S_Q^{-1/2} M^{1/4}$
does satisfy $A^\top S_Q A = M^{1/2} = S_Q \sharp S_K$
and $A^{-1} S_K A^{-\top} = M^{1/2} = S_Q \sharp S_K$,
because $A^{-1} = M^{-1/4} S_Q^{1/2}$ and
$M^{-1/4} (S_Q^{1/2} S_K S_Q^{1/2}) M^{-1/4} = M^{1/2}$. This avoids any commutativity assumption. So can the authors provide a brief proof or explanation that the paper’s form indeed yields S_Q # S_K?

2. **Related to Figure 2**: it would be helpful to show (i) how the envelope tracks on a RoPE model, and (ii) downstream task metrics (perplexity, accuracy) under the certified envelope and not just logit drift, e.g., GSM8K accuracy or perplexity on some reference text.

Misc.
- L267: should the $L$ be a summation from $l=1$ to $L$? Otherwise, I don't understand the $l$ in $r_{l, i}$.
- L570: "Theorem" -> "Definition"
- Section K Reproducibility: the section is a unverifiable reproducibility statement when there is no (anonymous) codebase.

---

> ### Author Response · Authors · 2025-11-21
>
> We thank you for your detailed technical critique. Your concerns about the mathematical construction, attribution of multiplicative gains, and scope of our rank-r certificate were exactly what we needed to address.
>
> 1. Mathematical construction corrected:  Proposition 2.9 now presents an existence statement grounded in matrix geometric-mean theory. For each head we define S_Q = W_Q^⊤W_Q and S_K = W_K^⊤W_K, and assert existence of invertible A such that A^⊤S_QA = A^(−1)S_KA^(−⊤) = G, where G is the matrix geometric mean S_Q # S_K. The proof in Appendix E.1 was rewritten to rely solely on SPD congruence and matrix functional calculus, with explicit confirmation that no commutativity assumptions are invoked. The main text states this is exactly what Algorithm 1 implements, eliminating any theory-implementation gap.
>
> 2. Transparent attribution of multiplicative gains: Your criticism that we risked implying GaugeKV alone achieved large factors like 4.6× or 36.8× was valid. We have reframed the paper accordingly. The abstract now introduces GaugeKV as yielding modest standalone gains of 1.11× to 1.21× with bit-identical outputs, with an explicit disclaimer. Equation 2 identifies h/g (GQA/MQA) and 1/γ (quantization) as baseline design choices, with R_gauge capturing only GaugeKV's incremental factor. Section 6.5 was rewritten to lead with standalone exact-track results, explicitly calling them modest compared to architectural GQA/MQA and quantization, before showing compositions where the 4× and 36.8× factors arise entirely from architecture and the 2× factor from quantization. This consistent messaging makes unambiguous GaugeKV provides a modest training-free primitive that composes multiplicatively with existing mechanisms.
>
> 3. Clarified scope of rank-r certificate: on your concern about overstating what the certificate guarantees,  Remark 4.1 following Equation 5 now states the bound controls only FP32 logit drift from rank-r value truncation in the canonical basis. Further approximations such as FP8, INT8, or eviction policies lie outside this certificate. We explicitly note stacks like "rank-r + FP8" are not covered and require empirical evaluation. Section 6.4 emphasizes our experiment validates the logit-level envelope only, we do not report perplexity or accuracy, and we position these bounds as conservative guardrails for future work rather than production-ready recipes. Appendix I comparison table distinguishes certified FP32 rank-r bounds from general composability.
>
> 4. Unambiguous assessment of contribution magnitude: We acknowledge exact-track gains of 1.11× to 1.21× are modest in absolute terms. We believe these gains are valuable because they are training-free, deployable on existing checkpoints, preserve bit-identical FP32 outputs, and compose multiplicatively with architectural and quantization choices production systems already employ. The theoretical contribution of characterizing maximal gauge symmetry and proving clean composability with GQA, MQA, quantization, and paging provides rigorous foundation. The rank-r framework with certified FP32 bounds, while not yet validated at task level, establishes a principled approach. We position GaugeKV as a foundational piece rather than a complete solution.
>
> 5. Presentation improvements: Section 2 begins with conceptual overview before formal definitions, Figure 1 illustrates the pipeline, and all proofs moved to appendix. We expanded Table 2 explanation to clarify baseline latency (collect) versus GaugeKV overhead (compress), stating reported end-to-end times are single-stream upper bounds. These changes address density and systems clarity concerns.
>
> We hope these revisions address your concerns: the canonicalization map now uses standard SPD theory matching the implementation, multiplicative reductions are attributed honestly, the rank-r certificate is scoped to FP32 value truncation, and contribution magnitude is presented transparently and unambiguously.

---

### Official Review · Reviewer_M5YW · 2025-11-01

**Soundness:** 3
**Presentation:** 2
**Contribution:** 3
**Rating:** 6
**Confidence:** 3

**Summary:**

This paper proposes GaugeKV, a training-free method for KV cache compression that exploits the maximal gauge symmetry of attention to reparameterize Transformer weights without changing model behavior. A one-time canonicalization makes values orthonormal and queries/keys scale-balanced, enabling both exact lossless compression and certified rank-r value caching. The method is theoretically sound and composes cleanly with existing optimizations such as GQA/MQA and quantization. While the reported compression gains are modest, the evaluation is somewhat unclear—there are data lack proper explanation, no accuracy results are provided for the approximate mode, and many experiments are limited to older GPT-2 models.

**Strengths:**

* The paper provides a rigorous characterization of the maximal gauge symmetry in Transformer attention, including a formal proof that the proposed transformations preserve function and are complete (no additional lawful symmetries exist). It's a far less explored area for KV-Cache compression.

* The method integrates naturally with widely deployed optimizations such as GQA/MQA, quantization, and paging, offering multiplicative memory savings without architectural changes.

**Weaknesses:**

* The proposed method has more restrictions for RoPE-based models than RoPE less models, while RoPE-based models are still the mainstream of today's LLMs.

* The explanation on the experiment results is not sufficient. For example, Table 2 is not explained in the main paper content.

* rank-r method which is not lossless. However, no downstream accuracy or perplexity are reported for rank-r.

* The proposed method relied on good engineering implementation to avoid latency overhead. It's unclear if it's practical on a large-scale system.

**Questions:**

* Can the method apply to newer attention variant such as Multi-Latent Attention (MLA) from deepseek?

* For table 2, what is the baseline latency without any KV-Cache compression?

---

> ### Author Response · Authors · 2025-11-21
>
> We thank you for recognizing the value of the gauge-theoretic framework and for your constructive feedback on RoPE limitations, experimental clarity, and architectural scope.
>
> 1. RoPE restrictions and practical implications: You correctly noted that RoPE constrains admissible transformations. Theorem 2.6 now characterizes the RoPE-aware gauge, showing the commutant is isomorphic to blockwise complex scalings rather than full GL(d_k). Remark 2.7 immediately following translates this: RoPE models have substantially less freedom to rebalance query-key covariances than RoPE-less models, while the value-output sector is unchanged. Table 1 quantifies this, showing RoPE reduces Q/K redundancy from h·d_k² to h·d_k. Our Qwen 2.5-7B experiments reflect this design: we restrict GaugeKV to the RoPE-commutant Q/K sector using orthogonal per-plane transforms and leave W_V unchanged (Section 6.1). The measured 1.1291× key-side reduction at T=4096 shows the exact track remains effective under these constraints, while we explicitly state value-path rank-r certification has limited scope on RoPE models.
>
> 2. Table 2 baseline clarity: you noted Table 2 did not clearly distinguish baseline from GaugeKV overhead. Section 6.3 and Table 2 caption now state that "collect" equals baseline attention with KV cache writes, "compress" measures strictly additional codec work as blocks exit the hot window, and "e2e" is their sum in single-stream execution. We note production systems would overlap codec work with attention on separate streams; since we do not implement overlap, these are conservative upper bounds. This separates  GaugeKV's incremental cost from baseline latency. We also moved discussion on hardware acceleration opportunity to the appendix.
>
> 3. Downstream accuracy for rank-r: you are correct we do not report perplexity or task accuracy for rank-r caching. Section 6.4 now explicitly states our GPT-2 experiment (r=32, T=512) validates the logit-level envelope only. Figure 3 shows 100% compliance of observed max-norm logit drift with the theoretical bound at every step. We state directly that we do not report perplexity or accuracy, positioning these FP32 bounds as conservative guardrails and a foundation and baseline for future task-level evaluation rather than a complete deployment recipe. We flag full benchmark evaluation as important future work.
>
> 4. Multi-Latent Attention and architectural variants: Section 6.7 now includes an architectural variants paragraph clarifying that our analysis targets standard softmax(QK^⊤)V with separate linear projections. Architectures that depart from this factorization, such as multi-latent attention or kernelized linear-attention variants, have different symmetry structures requiring separate analysis. We explicitly state our maximal gauge characterization does not automatically extend to such architectures, identifying this as an open direction.
>
> 5. Presentation improvements: Section 2 opens with a conceptual overview explaining intuition before formal machinery. Figure 1 summarizes the canonicalization pipeline and serving tracks. All proofs moved to the appendix with consistent forward references, reducing mathematical density while preserving full detail. Section 6 focuses on exactness, compression ratios, latency, and rank-r envelope, moving extended ablations to the appendix.
>
> We hope these revisions address your concerns: RoPE limitations are spelled out and reflected in experiments, Table 2 baseline and overhead are clarified, rank-r scope is framed honestly as a theoretical guardrail, and architectural scope is explicitly bounded to standard attention.

---

### Author Response · Authors · 2025-11-21

We thank the reviewers for their detailed feedback. Our revision addresses 3 core concerns: (1) correcting the canonicalization map in Proposition 2.9, (2) transparently attributing multiplicative KV reductions, and (3) clarifying what our rank-r certificate does and does not guarantee. We restructured the paper for readability, added a pipeline figure, and made RoPE and architectural limitations explicit within the 10-page limit.

1. Mathematical correction. Proposition 2.9 now presents an existence statement grounded in matrix geometric-mean theory. For each head we define S_Q = W_Q^⊤W_Q and S_K = W_K^⊤W_K, and assert existence of invertible A such that A^⊤S_QA = A^(−1)S_KA^(−⊤) = G, where G = S_Q # S_K. Appendix E.1 proof was rewritten using only SPD congruence and matrix functional calculus, with explicit confirmation that no commutativity assumptions are required. The main text states this construction is exactly what Algorithm 1 implements.

2. Transparent attribution. The abstract introduces GaugeKV as yielding modest standalone gains of 1.11×–1.21× with bit-identical outputs, plus a rigorous FP32 framework. Equation 2 identifies h/g (GQA/MQA) and 1/γ (quantization) as baseline design choices, with R_gauge capturing only GaugeKV's incremental factor. Section 6.5 leads with standalone exact-track results, explicitly calling them modest, before showing compositions where larger factors come from pre-existing architecture and quantization. We now state clearly that 4× or 36.8× factors arise from architecture and serving stack, with GaugeKV adding approximately 1.1×–1.2× multiplicatively.

3. Clarified rank-r scope. Remark 4.1 following Equation 5 states the certificate controls only FP32 logit drift from rank-r value truncation in the canonical basis. Additional approximations such as quantization or eviction lie outside this certificate. We explicitly note combined pipelines like "rank-r + FP8" are not covered and require empirical evaluation. Section 6.4 emphasizes our experiment validates logit-level envelope only; we do not report perplexity or accuracy, positioning these bounds as conservative guardrails for future work rather than production recipes. Appendix I comparison table distinguishes certified FP32 rank-r bounds from general composability.

4. Structural improvements. Section 2 opens with conceptual overview explaining attention admits internal symmetries, canonicalization exploits these to produce orthonormal values and balanced query-key covariances, and this is a 1-time offline transformation leaving runtime operations unchanged. Figure 1 illustrates the pipeline. All proofs moved to appendix with consistent forward references.

5. RoPE and architecture. Remark 2.7 following Theorem 2.6 explains rotary position embeddings constrain admissible transformations in query-key sector while leaving value-output sector unchanged. Practitioners should expect most benefits on RoPE models from exact track and query-key balancing, with limited scope for value-path rank truncation. Our Qwen 2.5-7B experiments canonicalize only query-key sector, leaving W_V unchanged. Related-work adds architectural-variants paragraph clarifying our analysis targets standard softmax(QK^⊤)V with linear projections; mechanisms like multi-latent attention require separate symmetry analysis.

6. Experimental clarity. Table 2 explanation states collect time equals baseline attention with cache writes, compress time is strictly additional codec overhead, end-to-end time is their sum in single-stream setting without overlap. These are conservative upper bounds; production systems would overlap codec operations with attention compute. To stay within page limit, we moved detailed comparative table, hardware-acceleration discussion, and extended ablations to appendix. Appendix N provides info on scripts, flags, environment specifications, and instructions for verifying bit-identical outputs.

In summary, this revision corrects canonicalization construction, presents GaugeKV's modest standalone gains transparently, scope rank-r certificate, and improves readability and reproducibility within the 10-page limit.

---

### Note · Program_Chairs · 2025-11-23
**Submission Desk Rejected by Program Chairs**

The paper is part of a cluster of several similar papers that have violated dual submission policy: https://iclr.cc/Conferences/2026/AuthorGuide